# Design of a Multichannel Pulser/Receiver and Optimized Damping Resistor for High-Frequency Transducer Applied to SAM System

Ngoc Thang Bui [1], Thi My Tien Nguyen [2], Tran Thanh Nam Dinh [1], Quoc Cuong Bui [1], Tan Hung Vo [1], Duc Tri Phan [1], Sumin Park [1], Jaeyeop Choi [1], Yeon-Hee Kang [3], Byung-Gak Kim [4,*] and Junghwan Oh [1,5,6,*]

[1] Industry 4.0 Convergence Bionics Engineering, Pukyong National University, Busan 48513, Korea; buingocthang1984@gmail.com (N.T.B.); thanhnam110793@gmail.com (T.T.N.D.); buicuong414@gmail.com (Q.C.B.); tanhung0506@gmail.com (T.H.V.); phanductribkhcm@gmail.com (D.T.P.); suminp0309@gmail.com (S.P.); eve1502@naver.com (J.C.)

[2] Department D of Pediatrics at the Hospital for Tropical Diseases, 764 Vo Van Kiet Street, Ward 1, District 5, Ho Chi Minh City 700000, Vietnam; bsmytien88@gmail.com

[3] (BK21 Plus) Marine-Integrated Biomedical Technology Program, Pukyong National University, Busan 48513, Korea; kynnuclear@naver.com

[4] College of Future Convergence, Pukyong National University, Busan 48513, Korea

[5] Department of Biomedical Engineering, Pukyong National University, Busan 48513, Korea

[6] Ohlabs Corporation, Busan 48513, Korea

\* Correspondence: bgkim@pknu.ac.kr (B.-G.K.); jungoh@pknu.ac.kr (J.O.); Tel.: +82-51-629-5771 (J.O.)

**Abstract:** Scanning acoustic microcopy (SAM) is widely used in biomedical and industrial applications in dermatology, ophthalmology, intravascular imaging, and small animal images, owing to SAM's ability to photograph small structures with a good spatial resolution. One of the most important devices of this system is the pulser/receiver (P/R) (PRN-300, Ohlabs Corporation, Nam-gu Busan, Republic of Korea), which generates pulses to trigger a high-frequency transducer. This article presents the design of a pulse generator to excite high-frequency transducers with four channels. The characteristics of the pulses, such as time and frequency, can be reconfigured by using a high-speed field programmable gate array (FPGA). The configuration software was developed for communicating with the P/R device via a USB connector for easy, feasible pulse selection and real-time pulse management. Besides that, during the design and implementation of the hardware, we optimized the damping resistor value to reduce the overshoot and undershoot part of the signal, ensuring the best effect on the transducer signal. The test results show that unipolar pulses worked with transducers with frequencies over 100 MHz. The SAM systems can work simultaneously with multiple transducers, and the resulting images have different resolutions of regions.

**Keywords:** scanning acoustic microscopy (SAM); high-frequency transducer; pulse generator; high-speed PCB design

## 1. Introduction

In recent years, the scanning acoustic microscopy (SAM) system has been widely used in the field of nondestructive testing to detect defects such as voids inside rigid structures [1–4]. The SAM system uses high-frequency ultrasonic signals to restore images of the internal structure of the object under examination [5,6]. A high-frequency ultrasound (>20 MHz) allows small structures to be captured because of its good spatial resolution in the order of tens of micrometers. The SAM system has been

used in many different fields, in which succession must be followed. Burak Altun et al. used the SAM system to analyze the impedance measurement of tissue. Their research plays an important role in ensuring the quality of ultrasound diagnostics [7]. Ryo Nagaoka et al. have developed a new calculation method for the SAM system to measure objects with a thickness of 1.95 um with an accuracy of 0.0016% [2]. C. Uhrenfeldt et al. adopted SAM to evaluate power electronic components with a nondestructive testing (NDT) method [8]. L. Pitta Bauremann et al. used SAM to identify battery defects [9]. M. Hertl et al. used SAM to scan interior images of electronic components [10]. The complete SAM system includes five main parts: a transducer, a pulser/receiver, a digitizer, a motion controller and image processing [1]. For high-frequency transducers with small focal zones, small dimensions can be detected in the structure to be tested [10]. Therefore, the ultrasonic transducer strategy, which converts electrical energy into high-frequency sound waves, is an important consideration for optimizing system performance and achieving high image quality [2,7].

There have been many studies on how to generate a signal to the excitation transducer [11]. These studies mainly focused on generating arbitrary waveforms to the excitation transducer to increase the signal-to-noise ratio (SNR) and increase the resolution of the resulting images [11–15]. However, these studies have not focused on deploying a complete device for nondestructive testing (NDT) and SAM systems. In this study, we will discuss the complete pulser/receiver (P/R) device used in the SAM system and the methods for increasing the scanning speed of the SAM system [12,16–20].

There are usually two types of pulses used to excite transducers: unipolar and bipolar pulses [2,21,22]. The bipolar pulses carry a high amount of energy, but the bandwidth is small [23,24]. The bipolar pulse is often used with applications that use low-frequency transducers [2,7,9,25]. Unipolar pulses with higher bandwidths bring smaller power, suitable for applications using high-frequency transducers. Therefore, unipolar pulses can produce a vertical resolution (interlayer resolution) better than that of bipolar pulses [9,26]. In this study, we will present the design of a unipolar P/R device to stimulate the transducer for imaging applications [27]. The programed P/R is fully compatible with SAM systems and many transducers with different operating frequencies. Typically, traditional SAM systems use one transducer, but our P/R equipment is designed to be used simultaneously with four transducers. Thus, the scanning speed of the new SAM system is four times that of a traditional SAM system.

One of the most critical problems of P/R devices is the generation of unipolar pulses with high voltages and small duty cycles (<10 ns) [8]. There are two steps to creating this pulse. The first step is using a field programmable gate array (FPGA) to generate pulses with small duty cycles and low voltage amplitudes. The second step is using this pulse to excite a high-frequency Metal-Oxide Semiconductor Field-Effect Transistor (MOSFET) in order to generate high voltage signals that trigger the transducer [26]. However, this pulse is very difficult to perform and is often directly affected by elements on the printed circuit board (PCB). High frequency signals on the PCB often include overshooting and undershooting parts. Therefore, the design of the PCB and P/R equipment should be optimized to achieve the best signal [28,29].

This paper makes two main contributions. We presented the structure and components of a complete P/R device with high flexibility that can easily be integrated into many SAM systems with different transducer types. Secondly, we presented a technique for using damping resistors to optimize unipolar pulses in order to excite high-frequency transducers, achieving a high signal-to-noise ratio. The PCB design for these signal lines is also presented in detail. The P/R device design has four channels, which makes it easy to deploy a SAM system with four transducers to reduce the scan time of a sample. This is one of the highlights when compared with similar products like the UT320 (UTEX Scientific Instruments Inc., Mississauga, ON, Canada) [30] (only one channel) or the DPR500 (Imaginant Inc., Pittsford, NY, USA) [22] (two channels). On the other hand, the P/R device can be expanded with eight channels.

The rest of this paper is organized as follows. Section 2 is a description of the completed design of the P/R device. The implementation of the hardware and code is introduced in Section 3. The detailed

setup for testing and the results are presented in Section 4. Finally, a concluding remark is given in Section 5.

## 2. Materials and Methods

A description of the main modules of the P/R device is presented in Figure 1. The FPGA (XC6LSX9-2TQG144C, Premier Farnell, Leeds, UK) [31] and Pulser STHV800 (LIFE.AUGMENTED., Monza, Italy) [32] chips were responsible for generating unipolar pulses to excite the high-frequency transducers. The PCB for this design required advanced technology of a high quality. The dsPIC33EP64GS508 [33] chip was responsible for communicating with the PC via the USB2COM (TITAN Electronics Inc., Chicago, IL, USA) interface, both to receive commands from the PC and to control the FPGA chip through 8-bit communication. Besides that, the dsPIC33EP64GS508 chip also selected the gain of the amplifier module. The trigger module was responsible for synchronizing signals with external signal sources or connecting to the motor encoder in the SAM system. We proposed a complete P/R device, which is shown in Figure 1.

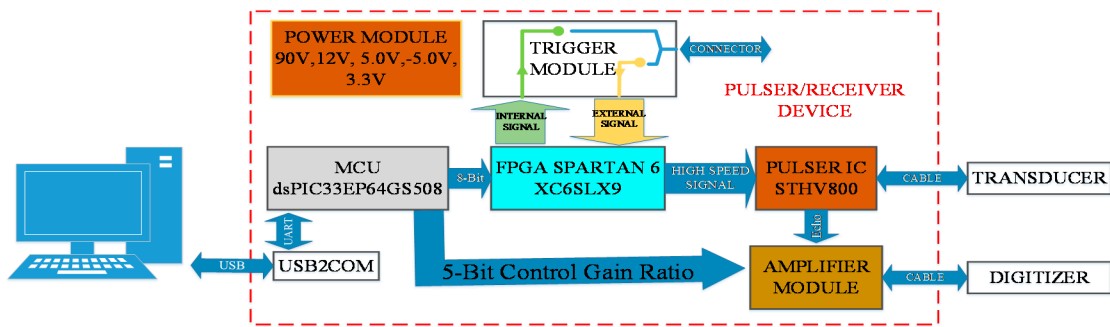

**Figure 1.** Block diagram of the pulser/receiver (P/R) device.

### 2.1. Configuration Software

Figure 2a describes the interface software for configuring the P/R parameters. The communication protocol between the P/R device and the software is described in Figure 2b. This is a fairly simple protocol which is used in a number of devices of the same type. The software was implemented based on Qt Creator 4.9 and compiled in C ++. Table 1 describes the functions of the configurational software for each channel of the P/R device.

**Table 1.** Functions of each channel of P/R software configuration.

| Items | Function's Name | Descriptions |
|:---:|:---:|:---|
| 1 | Enable/Disable | Enable or disable pulse signal out. |
| 2 | Trigger Source | Select if trigger sources are internal or external. |
| 3 | Mode | Select the mode of the device. There are two modes:<br>1. Pulse/echo mode: the device creates a pulse signal and receives the echo signal on the same channel.<br>2. Through mode: the device only receives an echo signal. |
| 4 | Trigger Repeat Frequency | Only for the internal trigger source, the device can select to repeat the frequency of the internal trigger (frequency from 0.1 kHz to 50 kHz). |
| 5 | Select Gain | Select the gain of the amplifier module (gain from 9 dB to 40 dB). |
| 6 | Voltage Out | Select the voltage out of the pulse signal (voltage out from 40 V to 90 V). |

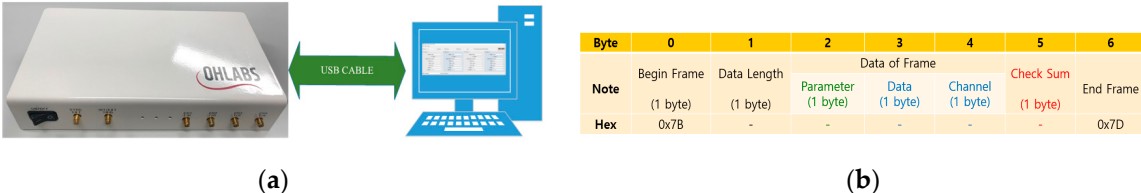

|  | (a) |  | (b) |

**Figure 2.** (**a**) Configurable parameters of the P/R device and (**b**) the interface protocol.

## 2.2. Hardware Design

Hardware design was one of the most important parts of our research. The hardware was implemented with Altinum Designer v.19 (Altium, San Diego, CA, USA) and was divided into two parts: schematic design and PCB layout. In this hardware design, we focused on a high-speed design for signal communication between the FPGA chip and the STHV800 chip, as well as the analog signal received from the transducer to be entered into the digitizer device. Besides that, the PCB layout is presented in detail for each module, including the designs of the pulser/receiver module, trigger module, and amplifier module.

## 2.3. Solution for Design the Pulser/Receiver Module

The STHV800 had eight independent channels. The device comprised the controller logic interface circuit, level translator, MOSFET driver, noise-blocking diodes, and high-power P and N channel MOSFETs as the output stage for each channel. The main issues during the PCB design process were achieving capacitance values that ensured good filtering and effective separation between the low voltage inputs and high voltage switching signals.

Figure 3 describes the hardware design to control a channel of the STHV800. Controlling a channel required two control signals from the FPGA, both of which required a high-speed circuit design (the duty part of the signal will determine the bandwidth of the P/R device). Besides that, the signal from the pin name XDCR was directly connected to the transducer to trigger the transducer with high voltage, then also receive an echo signal from the transducer. The signal from the pin name LVOUT was connected to the digitizer to analyze the transducer's echo signal. The signal from the XDCR and LVOUT pins was designed to meet the high-speed analog signals [32].

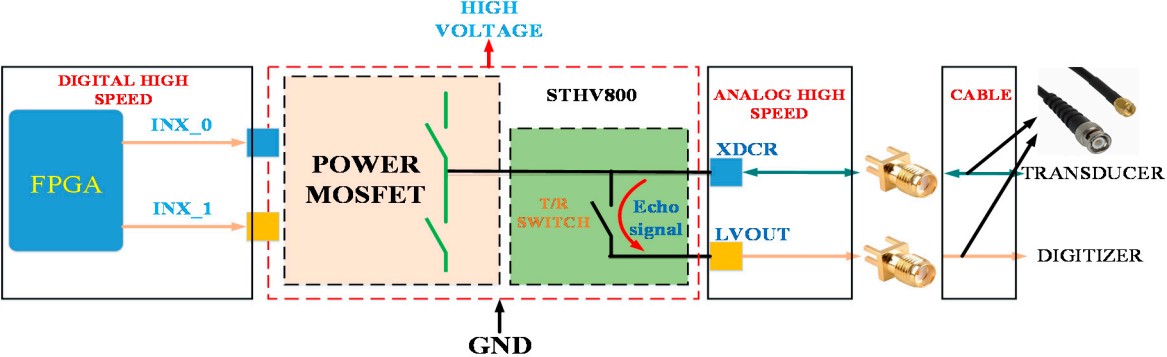

**Figure 3.** STHV800 single-channel block diagram.

## 2.4. Solution for Design the Trigger Module

The trigger module was capable of generating a square pulse signal with a frequency from 100 Hz to 50 kHz and receiving another square pulse signal with a frequency of up to 100 kHz. The trigger is an important module, significant in determining the transducer's focus point (internal trigger). The external trigger module could receive signals from the motor encoder and then create another trigger signal for the pulser and the digitizer, beginning the data receiving cycle. The operation of this module is described in Figure 4.

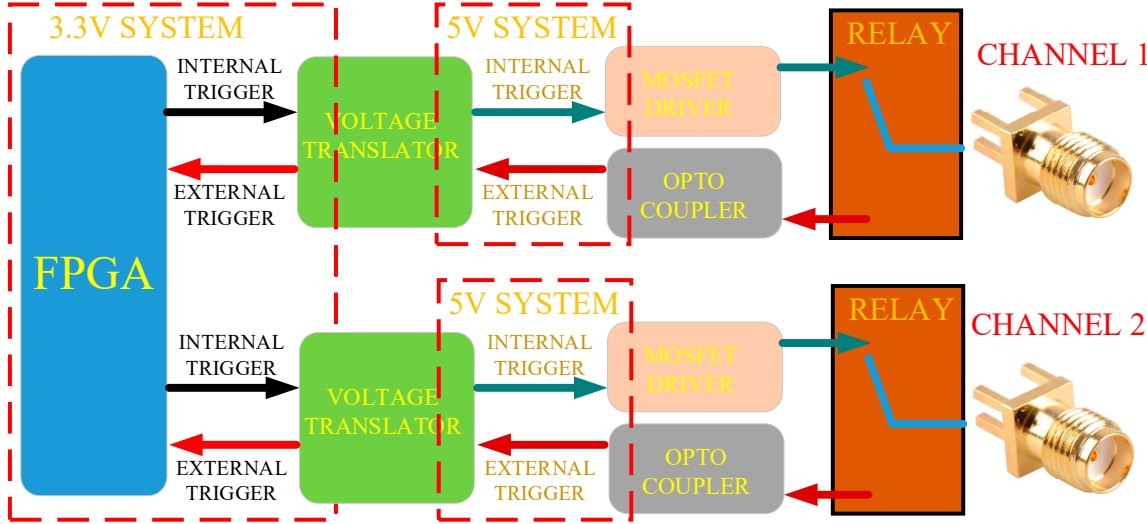

**Figure 4.** Block diagram of the trigger module.

Our design consisted of two of the same trigger channels, but they could work independently of each other. The connection of the trigger signal to the excitation channel of the STHV800 chip was configured using software.

### 2.5. Solution for Design the Amplifier Module

In this study, we designed a P/R device with four channels that could be operated independently. The signal from these channels was the input signal of the digitizer device used for analysis, so adjusting the amplitude of the signal was important in creating uniformity between channels. The gain amplifier of the channels could be adjusted from 9 dB to 40 dB, which meant we could use many different transducers at the same time.

The echo signal received from the transducer had a relatively small amplitude, so it was necessary to design the amplifier module to amplify the signal before entering it into the analyzer. In Figure 5, we show the parameters of the components that amplified the amplifier and the hardware after the design was completed [34].

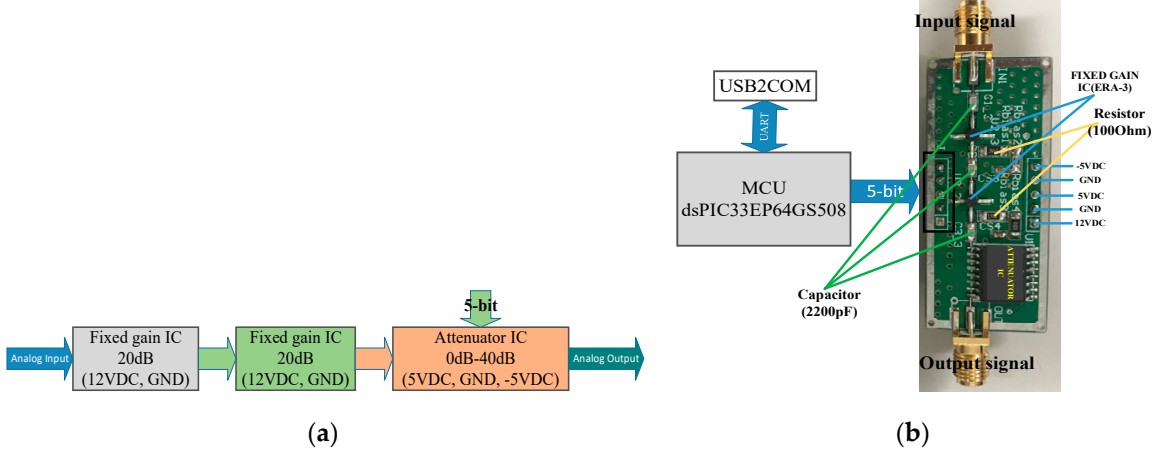

(**a**)                           (**b**)

**Figure 5.** Design of the amplifier module. (**a**) A block diagram of the amplifier module, and (**b**) the real amplifier module.

The gain amplifier was selected from the software. The dsPIC33 chip received the command from the software and then controlled the amplifier module's gain with a 5-bit resolution.

### 2.6. Unipolar Pulser Generator Analysis

High-voltage unipolar pulses are widely used in ultrasound applications because of their large bandwidth. A unipolar pulse signal is described in Figure 6. The shape of this pulse is analyzed into 3 parts. The first part consists of logic level 0 (rising edge), the second part is the time when the logic level is highest, and the last part is the overshot of the signal (falling time). For high-frequency transducer stimulation, signals require a small duty to click, thus reducing the rising time and falling time of the signal. In order to reduce these two parts of the signal, the impedance must be matched between the source and the load. At the same time, the signals need to be similar in length.

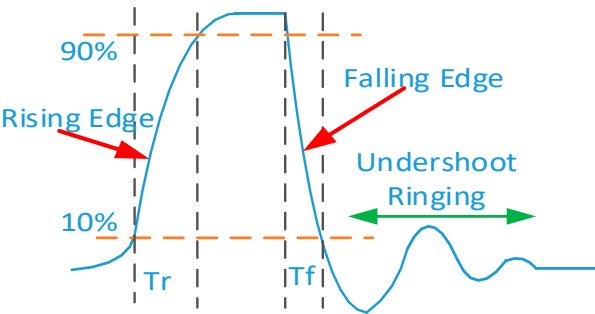

**Figure 6.** Characteristics of the excitation pulser for the transducer.

In this study, we needed to generate a unipolar pulse with a duty of less than 10 ns to trigger the transducer. This unipolar pulse was generated from the FPGA, then entered the input of the STHV800 chip to raise the voltage and trigger the transducer. The rising time and falling time components are described in Figure 6. To achieve such a signal, we needed to solve four major problems in the PCB design: signal reflection, signal noise, signal crosstalk and signal timing [28,35,36]. To solve these problems, based on the design of the PCB, we needed to ensure the proper values of the following factors: impedance, matching and spacing. The undershoot ringing in particular was solved by selecting damping impedance values suitable for each frequency of the transducer [37]. Reducing the undershoot ringing was essential for raising the bandwidth of the P/R device.

### 2.6.1. Impedance Controlled Routing

Stack layer design is one of the most important and effective methods in controlling the impedance of printed circuit lines in a PCB's design. Figure 7 depicts the impedance and propagation delay with two different types of circuit lines, which are narrow microstrips ($1 \leq \varepsilon_r \leq 15$ and $0.1 < w/h < 2.0$) and narrow striplines ($w/h < 0.35$ and $t/h < 0.25$).

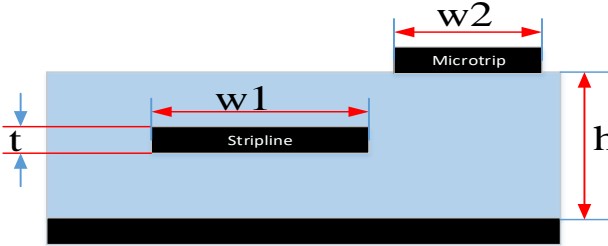

**Figure 7.** Characteristic impedance and propagation delay of a printed circuit board (PCB) trace.

Microstrip layout impedance and propagation delay can be expressed as

$$Z_0 = \frac{87}{\sqrt{\varepsilon_r + 1.41}} ln\left(\frac{5.98h}{0.8w + t}\right) \tag{1}$$

$$T_{PD} = 85 * \sqrt{0.475\varepsilon_r + 0.67} \tag{2}$$

Stripline layout impedance and propagation delay ca be expressed as

$$Z_0 = \frac{60}{\sqrt{\varepsilon_r}} ln\left(\frac{1.9h}{0.8w + t}\right) \tag{3}$$

$$T_{PD} = 85 * \sqrt{\varepsilon_r} \tag{4}$$

Another method to reduce the reflection of signals on the transmission line uses termination techniques. There are two broad kinds: series termination and parallel termination. Thevenin termination is one of the parallel terminations, shown in Figure 8. The resistance formula of a Thevenin termination is R1 + $Z_0$ = R2. Proper termination can reduce overshoots, undershoots and ringing, as well as the amplitude of crosstalk [38].

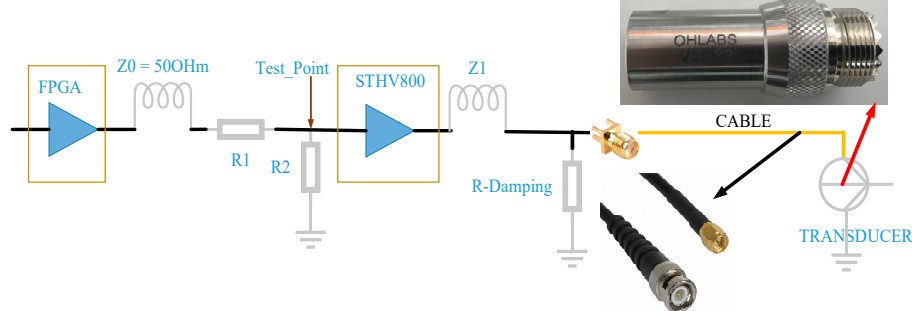

**Figure 8.** A Thevenin termination.

To analyze the design, a common rule of thumb is the one-third rise time rule, which states that if the route is more than one third of a rise time long, reflections can occur.

The one-third rise time rule states that the speed at which electrical energy can travel along a route is known as the propagation velocity and can be defined as

$$V_p = \frac{C}{\sqrt{\varepsilon_r}} \tag{5}$$

where $V_p$ is the propagation velocity, $C$ is the speed of light (299.792458 mm/ns) and $\varepsilon_r$ is the dielectric constant.

Applying the one-third rise time rule, the transmission line effects will begin when

$$L_R \geq \frac{T_R}{3} \times V_p \tag{6}$$

where $L_R$ is the length of a route and $T_R$ is the signal Rise time.

Assume a simple wire require the following calculation:

$$L_{max} \leq \frac{T_R}{2T_{DP}} \text{ or } delay_{max} \leq \frac{T_R}{2} \tag{7}$$

where $T_R$ is the signal rise time, $T_{DP}$ is the delay per unit of length, $L_{max}$ is the length of the wire and $delay_{max}$ is the wire delay, which is equal to $T_{DP} * L_{max}$.

The PCB design usually applies two rules to reduce the noise of the circuit board: the 20H rule and the 3W rule. Applying the 20H and 3W rules can reduce the circuit field's radiation by 70%. The 3W rule represents the approximate 70% flux boundary at logic currents [39]. In this design, to reduce the impact of crosstalk noise on the pulse signals from the FPGA with the STHV800, we applied the 3W rule. Table 2 describes the signal connection between the XC6XLX9 and the STHV800.

**Table 2.** Length and width of the pulse signal between the field programmable gate array (FPGA) and the pulse component.

| Items | XC6XLX9 PIN | STHV800 PIN | Length (Mil) | Width (Mil) |
|---|---|---|---|---|
| 1 | 8 | 28 | 1520 | 8 |
| 2 | 10 | 27 | 1520 | 8 |
| 3 | 12 | 26 | 1460 | 8 |
| 4 | 15 | 25 | 1460 | 8 |
| 5 | 17 | 24 | 1400 | 8 |
| 6 | 22 | 23 | 1400 | 8 |
| 7 | 24 | 22 | 1340 | 8 |
| 8 | 27 | 21 | 1340 | 8 |

2.6.2. Power Integrity Design

Using the signal integrity (SI) analysis method resulted in signal lines on the PCB to ensure signal integrity. However, energy integrity was a major issue that needed to be considered, especially for high-speed PCB types. Power integrity (PI) analysis indicated that interference from the power supplies greatly influenced signals in the high-speed systems. The noise voltage caused by the transient current made the supplying power continuous and even affected the normal work of the high-speed system. At present, PI issues have two solutions: optimizing the layer stack structure of the PCB and placing decoupling capacitors on the PCB.

In addition, we applied proper bypass capacitors and separation techniques to improve the overall power supply SI. A series of capacitors should be used to eliminate the effects of source voltage interference with the two main chips (FPGA and dsPIC33EP64GS508) in this design. Small decoupling capacitors with low current inductances could provide fast current to high-frequency converters. The average bypass capacitor could filter out noise, preventing them from entering the IC. Bulk capacitors continued to provide current after the high-frequency capacitors exhausted their energy storage. The application of these capacitors to the design is described in Figure 9.

A good layer stack structure may be the best solution for most SI and electromagnetic compatibility (EMC) issues. Ground planes are an effective EMC shield, good image planes can be good return paths, and small loop areas reduce electromagnetic interference radiation. The closer the power plane is to the ground plane, the bigger the capacitance, and the lower the impedance, the better the reduction of switching noises.

In this design, we used the eight-layer PCB with the stackup, as described in Figure 10. With this stackup, the impedance of the signal classes was calculated and matched with an impedance of 50 Ω. In this stackup, the third layer (high voltage) and the fifth layer (low voltage) separated. The layout of the source for the STHV800 chip had been optimized by us. The second layer (GND1, mainly ground for the dsPIC33EP64GS508, FPGA, . . . ), fourth layer (GND2, mainly ground for the low voltage of the STHV800) and seventh layer (GND3, mainly ground for the high voltage of the STHV) were the plane layers. We separated these layers to limit interference from the source.

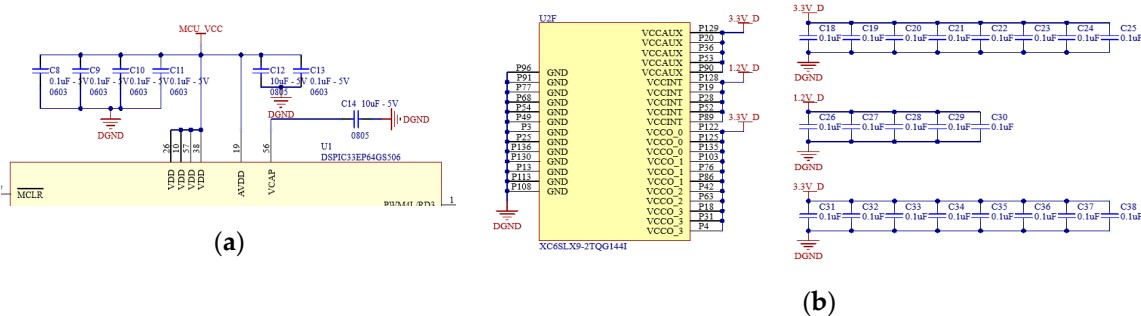

**Figure 9.** *Cont.*

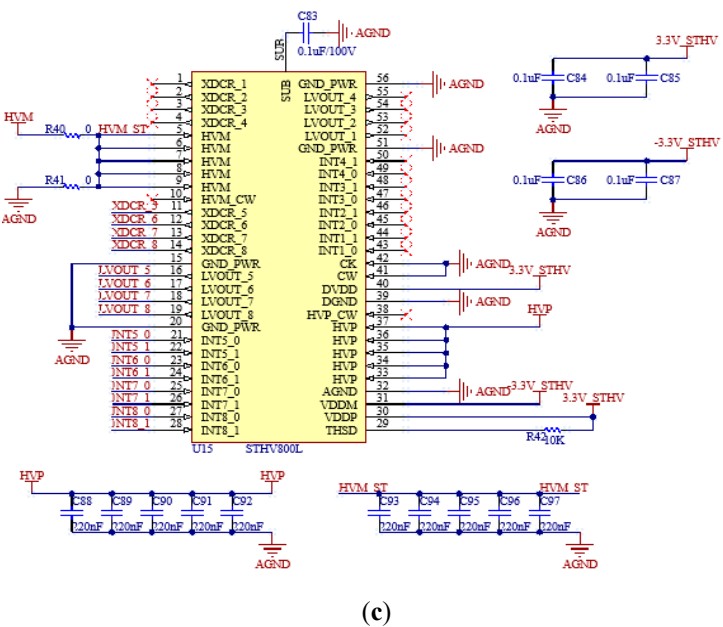

**(c)**

**Figure 9.** Design of the decoupling capacitor for the (**a**) dsPIC33EP, (**b**) XC6SLX9 and (**c**) STHV800.

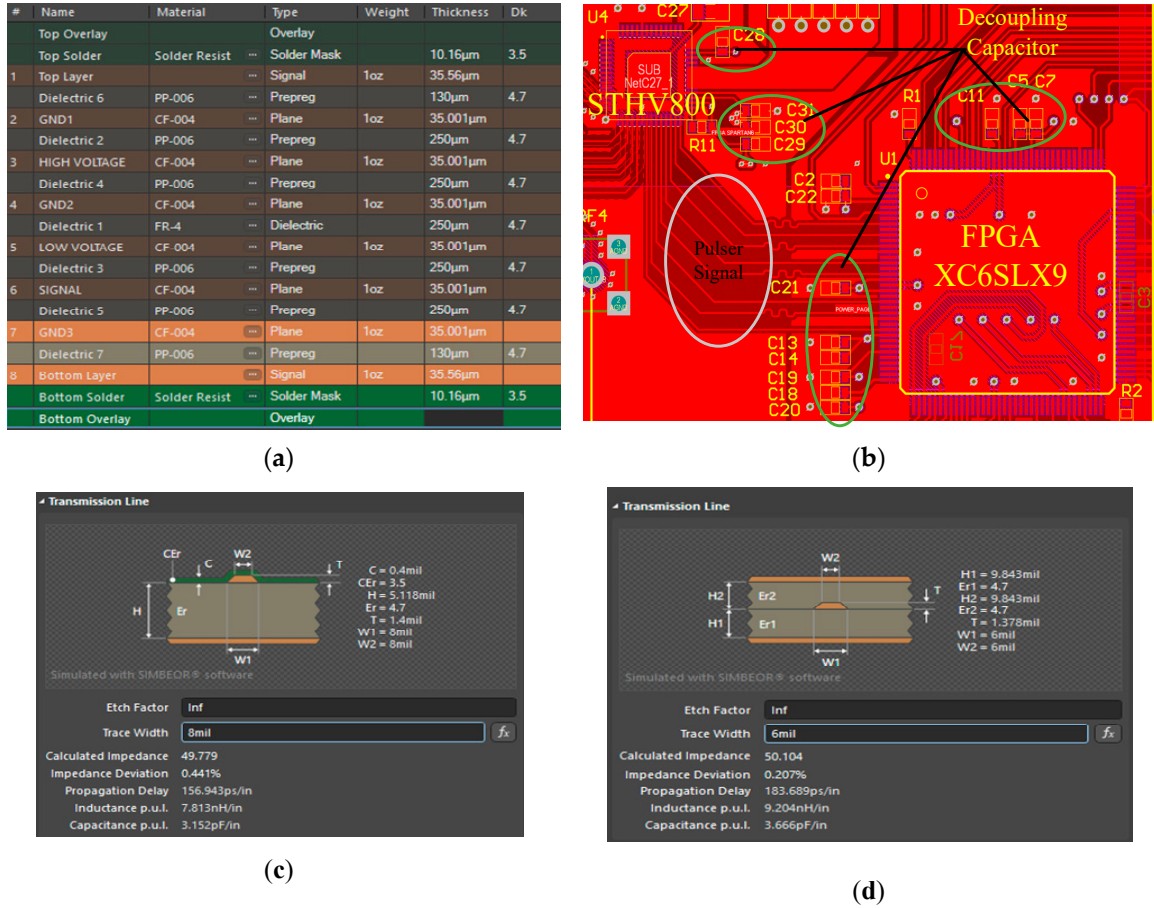

**Figure 10.** (**a**) PCB stackup for the STHV800 layout; (**b**) PCB layout with the decoupling capacitor and signal integrity (SI) technique; (**c**) impedance with the microstrips; and (**d**) impedance with striplines.

The layout design of the decoupling capacitors for the two main chips (FPGA and STHV800) are presented in Figure 10b, and the impedance of the microstrips and striplines are shown in Figure 10c,d.

### 2.6.3. Optimized Damping Resistor for High-Frequency Transducer

The techniques of matching impedance and PI are very effective with the onboard signals that help reduce overshoot and undershoot and optimize transmission power. The length of the cable between the transducer and the P/R device has not been determined (regarding the length of the cable and the impedance and frequency of transducer operation). As such, in this section, the damping resistance values need to be optimized for a different transducer type. Figure 8 shows the position of the damping resistor in the P/R device. However, the transducers do not match the impedance, so the use of damping resistors is important in improving the transducer's vertical resolution.

## 3. Results

### 3.1. Hardware Implementation

The hardware design of the P/R device is described in Figure 11. This hardware design consisted of four boards: the mainboard, the trigger board, the amplifier board and the high-voltage board. This design was implemented in Altinum Designer v.19. The software in which the trigger board, amplifier board and high-voltage board were designed had only two layers. The mainboard was designed with eight layers on this board, with applied SI and impedance matching techniques to generate pulser signals with appropriate duty.

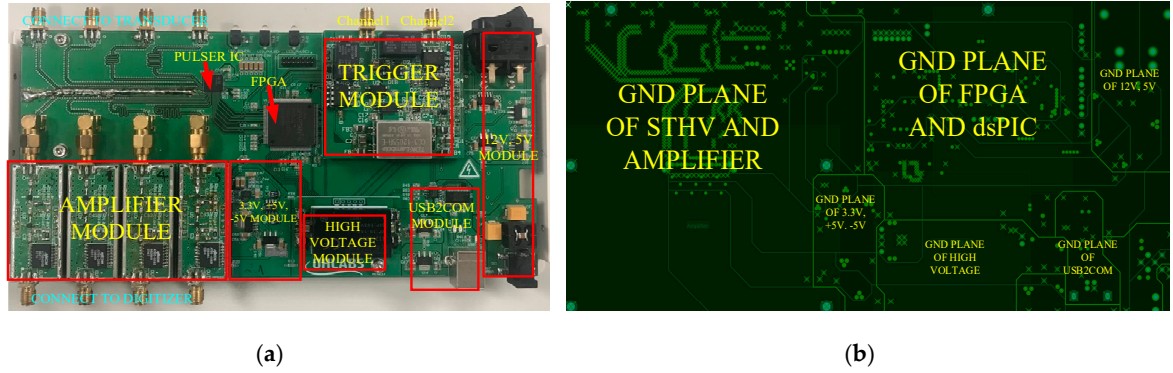

| (a) | (b) |

**Figure 11.** Hardware layout of the P/R device. (**a**) The real PCB after soldering components and (**b**) the GND plane of modules.

### 3.2. Solution Program the Code for the FPGA

The code implemented on the FPGA was implemented in ISE Design Suit 14.7, and it was divided into two parts. Part one used the intellectual property (IP) core (Clocking Wizard V3.6) to create the system clock for the system, with an input clock of 250 MHz, and the system clock generated was 300 MHz. This clock directly affected the signal to trigger the transducer. The second part was the code that performed pulse generation for the STHV800 chip and the internal or external trigger and interfaced with dsPIC33EP. The results of the clocking module are described in Figure 12.

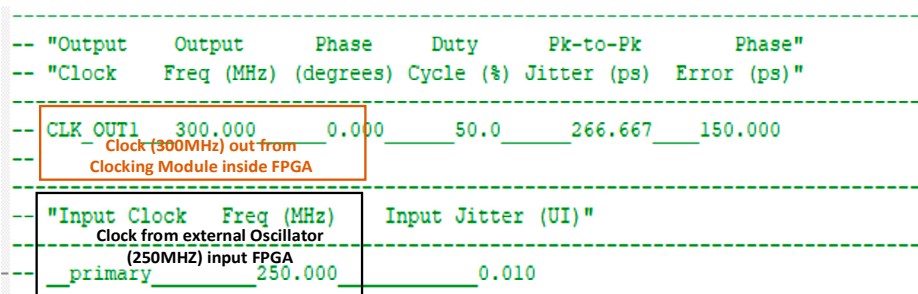

**Figure 12.** Results generated from the clocking module.

Figure 13 shows the program description for the FPGA chip implementation of two modules. Figure 13a describes the module for the internal or external trigger signals. Figure 13b describes the excitation generator module for the STHV800 chip. After completing the firmware for the FPGA, we tested and determined the latency of the system to be around 280 ns.

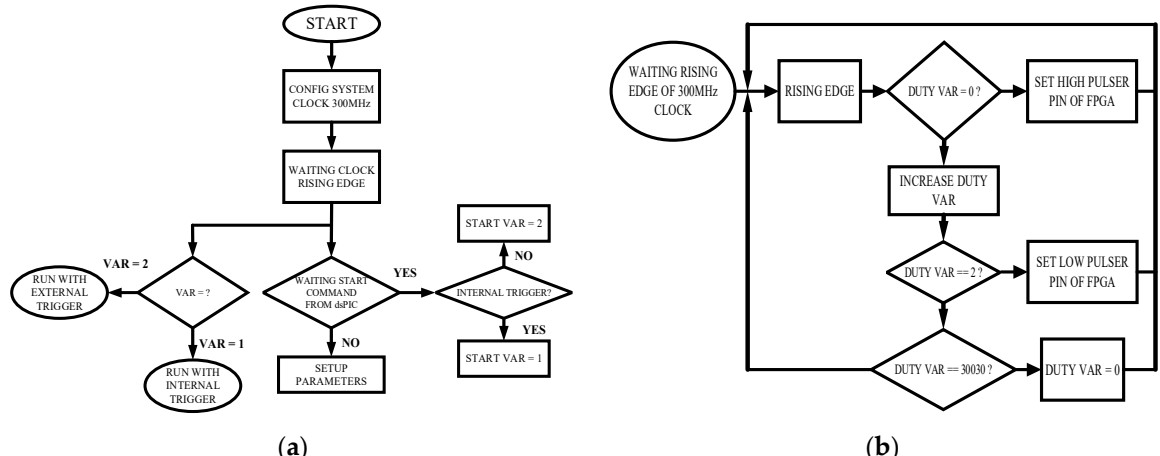

(a) (b)

**Figure 13.** Block diagram of the program inside the FPGA. (**a**) The main program; (**b**) The pulser program with an internal trigger = 10 kHz.

## 4. Implementation System and Results

After completing the hardware design, as well as acquiring the firmware and software needed for our system, we set up the SAM system to test the P/R device. The system model is described in Figure 14a, and the real model of the system is described in Figure 14b. The system test with the P/R equipment was divided into two parts. The first part was setting up and checking the signals that were designed and programmed with the transducer and the class sample. The second part tested the SAM system when scanning samples with the techniques applied above.

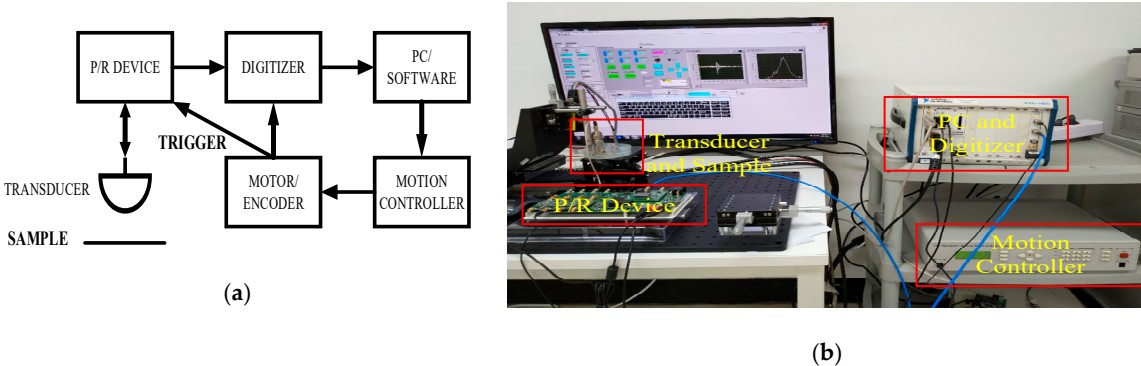

(b)

**Figure 14.** Set-up of a scanning acoustic microscopy (SAM) system with a P/R device. (**a**) A block diagram of the SAM system; (**b**) A real SAM system.

### 4.1. Unipolar Pulser Generator

As we mentioned above, the pulse optimization for the probe was done in two parts. The first part, performed on the included PCB design, ensured the integrity of the trigger signals from the FPGA to the STHV800. Since the transducer impulse had a duty of only a few nanoseconds, the rising time portion was also a value of a few nanoseconds. We applied techniques such as calculating the characteristic impedance of the circuit, as in Equations (1) and (3), and calculating the length of the printed circuit and used thenvenin resistors to reduce the reflection of the signal. (Altinum Designer software also

provide the signal integrity simulation feature to shorten the time and improve the efficiency of the design). Besides that, we placed decoupling capacitors near the FPGA and the STHV800 to reduce the noise interference when they were working. For the second part, we used a damping resistor for the signal from the STHV800 to the transducer. The value of the damping resistor had been determined by tests because the impedance of the transducers may have differed.

The trace width and resistor values used in the design are described in Table 3.

**Table 3.** Value of the resistor for Thevenin termination.

| Items | Signals | R1 (Ω) | R2 (Ω) | Length (Mil) |
|-------|---------|--------|--------|--------------|
| 1 | STHV_IN1_0 | 45 | 50 | 1520 |
| 2 | STHV_IN1_1 | 45 | 50 | 1520 |
| 3 | STHV_IN3_0 | 45 | 50 | 1520 |
| 4 | STHV_IN3_1 | 45 | 50 | 1520 |
| 5 | STHV_IN5_0 | 45 | 50 | 1520 |
| 6 | STHV_IN5_1 | 45 | 50 | 1520 |
| 7 | STHV_IN7_0 | 45 | 50 | 1520 |
| 8 | STHV_IN7_1 | 45 | 50 | 1520 |

In this test, we measured the unipolar pulse signal generated from the FPGA chip to excite the STHV800 chip and evaluated the effectiveness of the techniques used in the PCB design with this signal. As mentioned above, we used the schematic described in Figure 8 to design the signal from the FPGA to the STHV800 and set the signal of the excitation transducer 50 MHz. The measurement point is denoted as Test_Point. We used a KEYSIGHT oscilloscope, model MSO-X2024A, to monitor the signal.

After a thorough analysis and comparison of the unipolar pulse signals in Figure 15, we can see that this signal improved. The undershoot part of this signal had been reduced significantly, which was essential in increasing the transducer signal's bandwidth. In Figure 15b, we see clearly that the damping resistor effect reduced the signal amplitude, but at the same time, the ringing part of the signal was also clearly reduced. This had important implications for detecting different layers of the object to be scanned because for many samples with many layers, there are often many echo signals received, and creating the vertical resolution of the sample depends on these echo signals. In Figure 15a, we compare the signal before and after adding damping resistors, the signal from the FPGA to the STHV800 had reduced overshoot and ringing with damping resistor. We could easily change the pulse width by changing the variables in the FPGA with the software to make a signal suitable for different transducers.

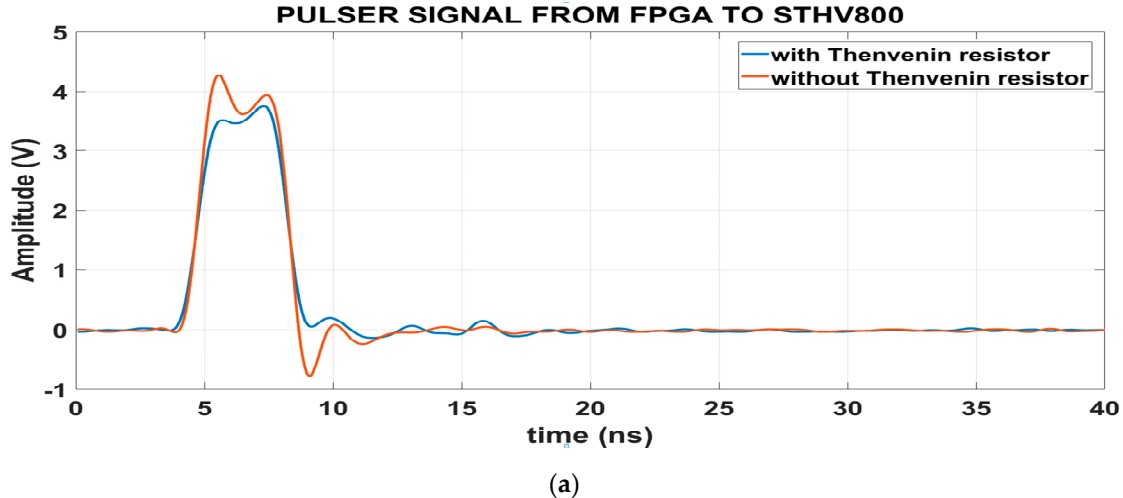

(a)

**Figure 15.** *Cont.*

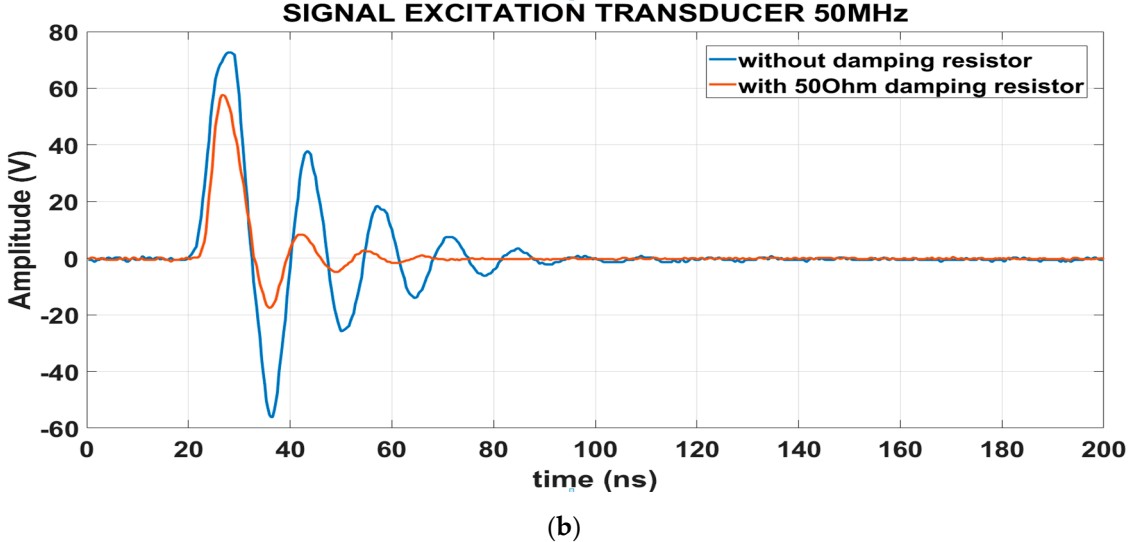

(**b**)

**Figure 15.** (**a**) The pulser signal from the FPGA to the STHV; (**b**) The signal excitation transducer.

### 4.2. Tested Background Noise of the P/R Device

In this test step, we measured the background noise amplitude of the P/R device with different gains from the amplifier module. The goal of this test was to check the P/R device's interference amplitude under normal operating conditions. These results also evaluated the effect of environmental interference on the device when combined with other components of the SAM system. The results are depicted in Figure 16. As the gain increased, the noise amplitude increased, but the maximum amplitude of the noise was 45 mV. Usually, these values are quite small relative to the amplitude of the echo signal. This result was important for identifying and eliminating interference from other devices that could affect the echo signal.

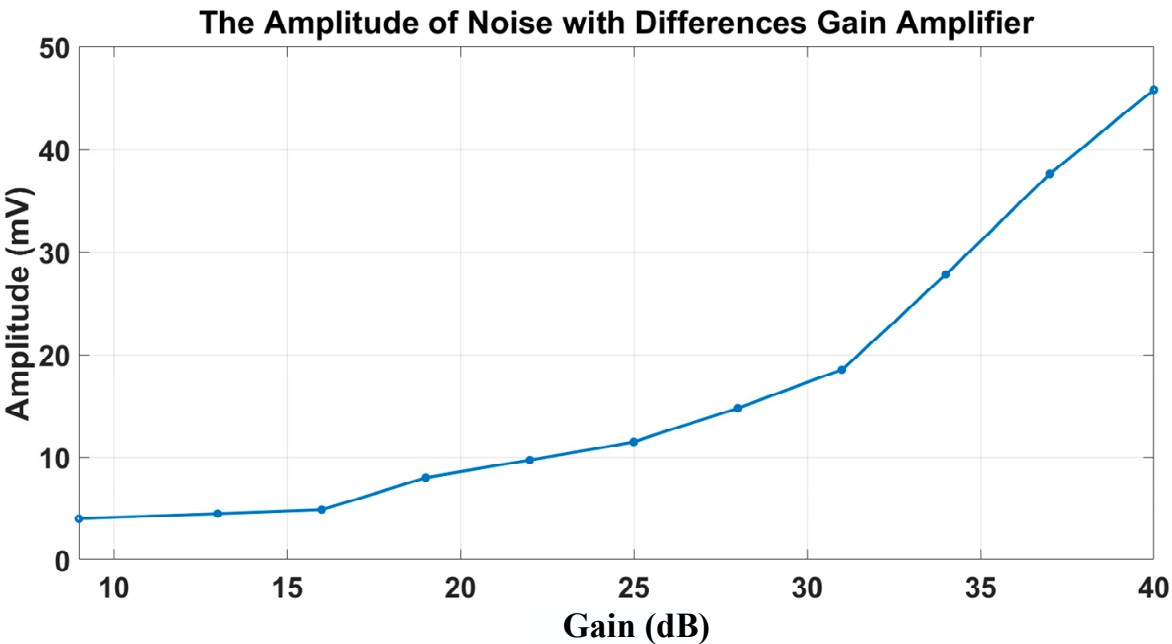

**Figure 16.** Amplitude of the background noise with differences in gain.

We ran tests with different echo signals. The maximum amplitude of the echo signal was about 3.5 V. The echo signals, after going through the amplifier module, had amplitudes higher than 3.5 V, but then the output signals were distorted.

### 4.3. Optimized Value of the Damping Resistor

As mentioned above, in this section, we will determine the damping resistor values best suited to the transducers of predetermined frequencies. We used the same experimental model as in Figure 15b. We replaced the PC and digitizer with an oscilloscope (MSO-X2024A) to observe the transducer's echo signal. This test result is presented in Table 4.

**Table 4.** Damping resistor value for different transducers.

| Frequency (MHz) | R-Damping (Ω) |
|---|---|
| 25 | 25 |
| 50 | 47 |
| 85 | 50 |

For the sample thickness, applications used ultrasound signals. The damping resistor was used to reduce the ringing portion of the first echo signal so that it was easy to detect the second echo signal. We chose for the gain of the amplifier module on the P/R device to be 40 dB and the excitation voltage to be 90 V.

As shown in Figure 17a, we describe a sample of steel which had a uniform thickness of 0.25 mm. In Figure 17b, we describe two echo feedback signals from the upper and lower sides of the sample to the transducer. In Figure 17c,d, we describe the echo signal measured on the oscilloscope of the 50 MHz transducer with and without a 47 Ω damping resistor.

Based on the echo signal described in Figure 17c,d we can see the function of the damping resistor when detecting echo signals close to each other. In Figure 17c, the transducer without an additional damping resistor made it difficult to identify the second echo signal because the ringing part of the first echo signal was too long. In Figure 17d, the transducer had a 47 Ω damping resistor added on the ringing part of the suppressed first echo signal, and it was easy to identify the second echo signal. This was because it depended on the characteristics of the transducer used in the experiment.

On the other hand, we computed the signal-to-noise ratio (SNR) of the signal depicted in Figure 17c,d. The results are shown in Table 5.

**Table 5.** Compared signal-to-noise ratio (SNR) of the 50 MHz transducer before and after applying the damping resistor.

| Items | SNR (First Echo) | SNR (Second Echo) |
|---|---|---|
| No Resistor | 5.96 | 1.36 |
| 47 Ω | 8.89 | 2.11 |

The results in Figure 17c,d and Table 5 show that the amplitude of the echo signal was reduced after applying the damping resistor. However, the damping resistor also suppressed a number of other reflected signals. Thus, this increased the SNR ratio of the echo signal. This also increased the quality of the B-mode images.

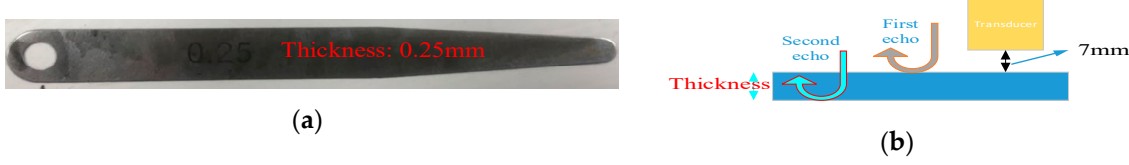

(**a**)  (**b**)

**Figure 17.** *Cont.*

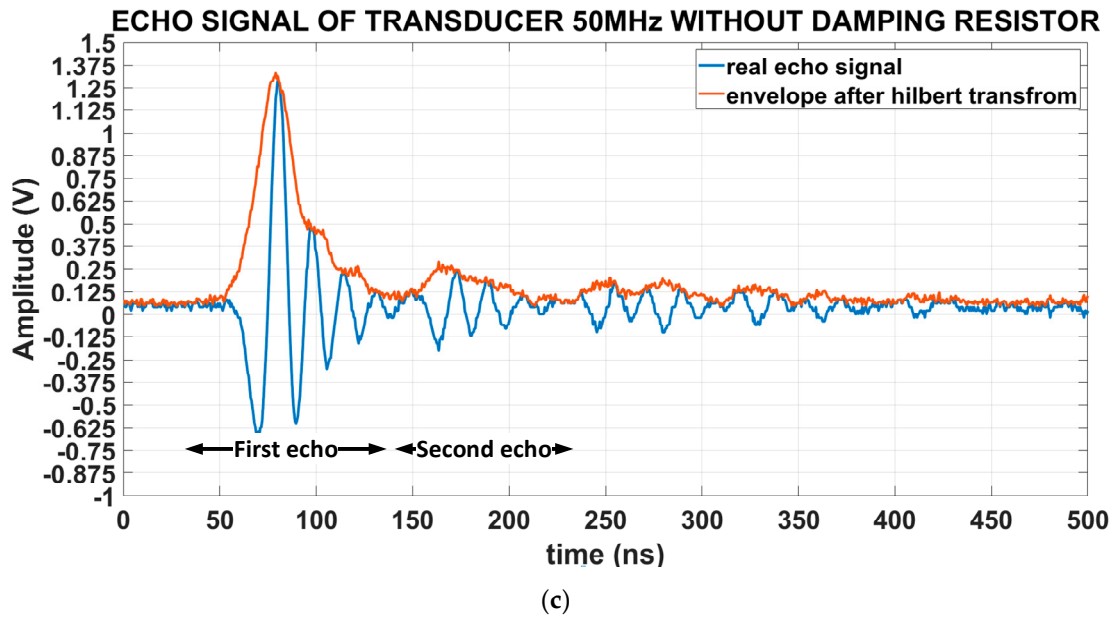

(c)

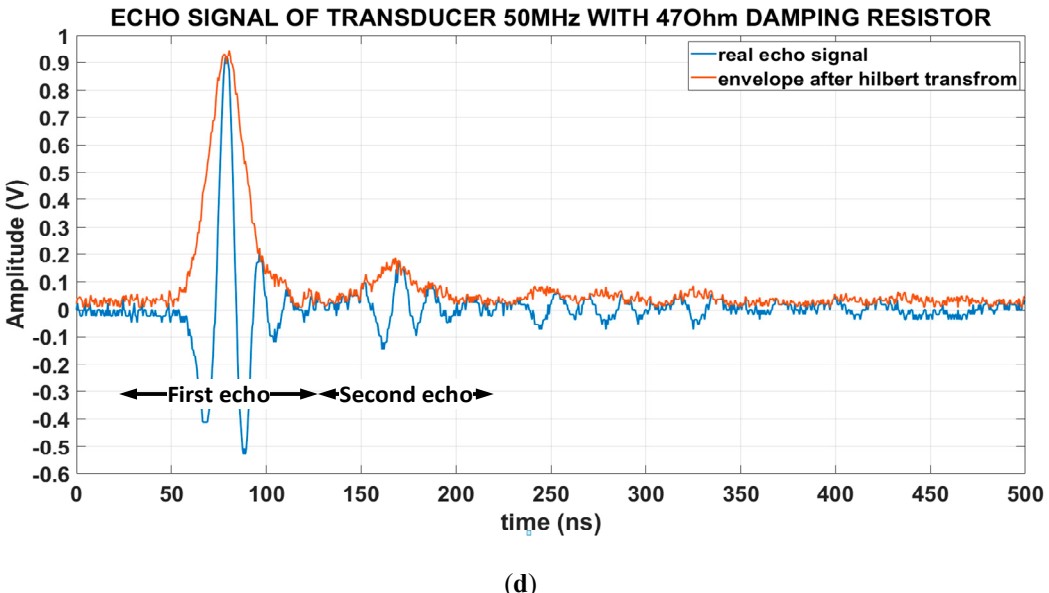

(d)

**Figure 17.** (**a**) The real steel sample with a thickness of 0.25 mm; (**b**) A model of the echo signals from the sample; (**c**) The echo signal of the 50 MHz transducer without a damping resistor; (**d**) The echo signal of the 50 MHz transducer with damping resistors of 47 Ω.

### 4.4. Testing with the SAM System

As described in Figure 15, the SAM system transmits and receives signals based on the signal from the motor encoder (this is also the trigger signal for the P/R and the digitizer). The SAM system consisted of three main devices: a PC and digitizer (NI PXIe-1062Q and NI PXI-5152 8-bit, 2 GS/s digitizer, National Instruments Corporation. Austin, TX, USA), a Newport Driver (Model ESP300, Newport Corporation. Irvine, CA, USA), and our P/R device. Therefore, the higher the pulse frequency from the encoder, the faster the sample scan time. In our design, the trigger module of a P/R device could receive pulses of up to 100 kHz.

In this test, we used an image acquisition system to test the signal with a sample of one coin. The parameters for setting up the scanning system are described in Figure 18 and Table 6.

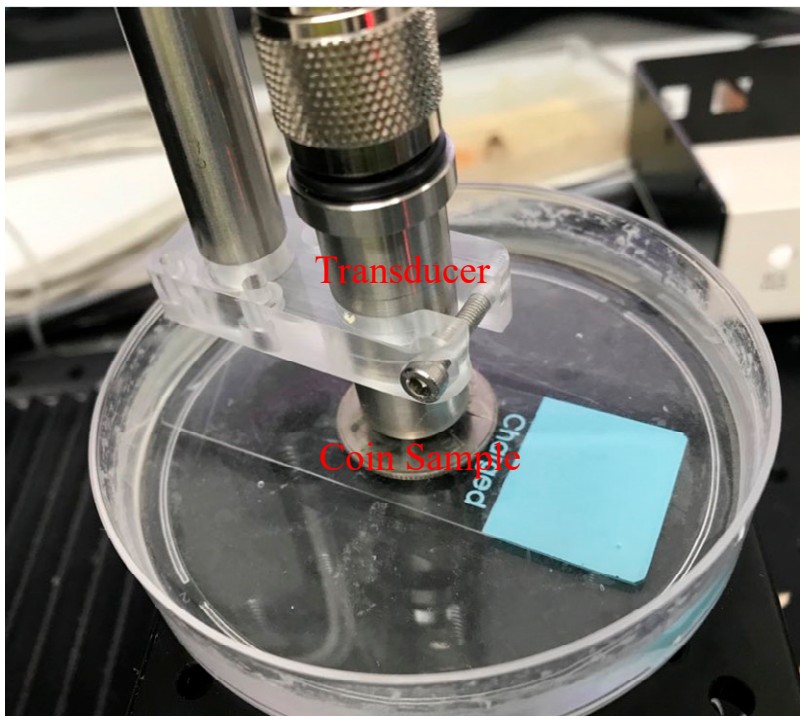

**Figure 18.** Set-up of the transducer with a coin sample.

**Table 6.** Parameters of the SAM system for coin scanning.

| Items | Transducer | R-Damping (Ω) | Step size | Number A-Scan | Number B-Scan |
|-------|-----------|---------------|-----------|---------------|---------------|
| 1 | 25 MHz | 25 | 100 μm | 500 | 500 |
| 2 | 85 MHz | 50 | 100 μm | 500 | 500 |

On the other hand, we conducted experiments to determine the SNR ratios of two different transducers. This test was done before the system's scan of the coin. The results are described in Table 7.

**Table 7.** Compared SNR of echo signal with and without damping resistor.

| Items | Transducer | SNR (no Damping Resistor) | SNR (with Damping Resistor) |
|-------|-----------|---------------------------|------------------------------|
| 1 | 25 MHz | 6.5 | 7.3 |
| 2 | 85 MHz | 7.3 | 9.1 |

In Figure 19, we described the system's final image with the use of damping resistors. The images obtained for systems using suitable damping resistors were of a high definition due to the improved SNR of the echo signal.

In Figure 19a, we scanned the surface of the coin and used a high-frequency transducer (85 MHz). High-frequency transducers with poor sample penetration are suitable for surface scanning applications. In Figure 19c, the inner layer of the Spartan-6 chip is described. We used a 25 MHz transducer to do this experiment. The ultrasonic waves generated by the 25 MHz transducer penetrate the object better than the 85 MHz transducer. Hence, the SAM system using the 25 MHz transducer obtained a better image than the SAM system with 85 MHz transducer.

After performing the coin scan, we calculated the contrast-to-noise ratio (CNR) of the obtained image. The results are described Table 8.

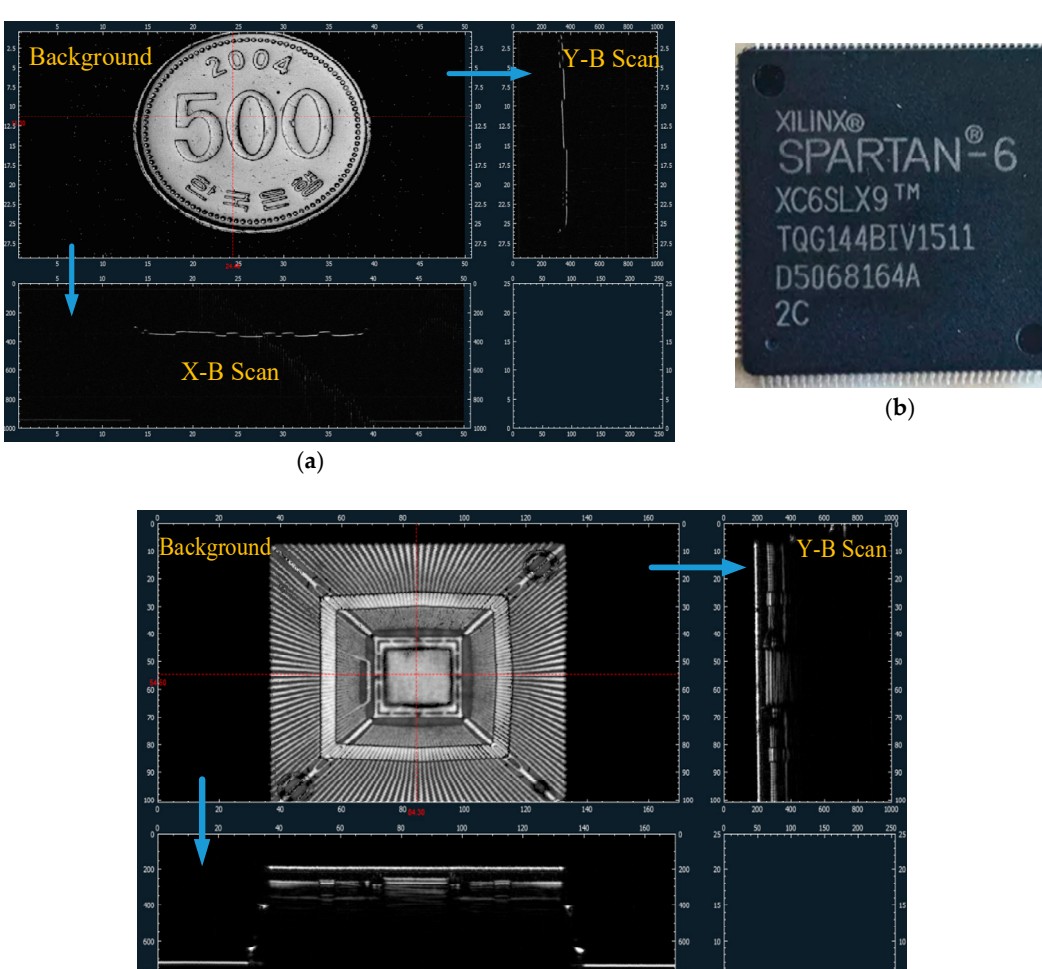

**Figure 19.** (**a**) Coin scanning image with an 85 MHz transducer; (**b**) Real image of the Spatarn-6 XC6SLX9-144; (**c**) Scanning image of the inside of the Spatarn-6 with a 25 MHz transducer.

**Table 8.** Compared contrast-to-noise ratios (CNRs) of images with a damping resistor.

| Items | Transducer | CNR (with Damping Resistor) |
|-------|------------|-----------------------------|
| 1 | 25 MHz | 8.2 |
| 2 | 85 MHz | 3.5 |

Based on the results in Tables 7 and 8, we can see that the damping resistor increased the SNR of the echo signal from the transducer. The CNR of the image was also improved after using damping resistors.

### 4.5. Scanning Sample with Four Transducers at the Same Time

One of the highlights of our design is its ability to operate with four transducers simultaneously. With this capability, the SAM system reduces the uptime by only a quarter compared with that of using the UT320 equipment. Moreover, the maximum trigger frequency of the UT320 was 20 KHz, compared with our P/R device's 50 KHz. That means the SAM system that worked with our equipment had the highest possible speed, which was 10 times faster than the SAM system using the UT320. Moreover, the highest amplitude of the echo signal from our P/R is 3.5 V, and the UT320 is 1 V. The echo

signal with high amplitude create high resolution for B-Mode images. Below is Table 9, comparing the main parameters.

**Table 9.** Compared main parameters of the UT320 [30] and the Ohlabs P/R.

| Items | Parameter | Ohlabs P/R | UT320 |
|---|---|---|---|
| 1 | Number of channels | 4 | 1 |
| 2 | External trigger | 50 KHz | 20 KHz |
| 3 | Max amplitude of echo | 3.5 V | 1 V |
| 4 | Weight | 1.2 Kg | 13.7 Kg |
| 5 | Size | $104 \times 482 \times 507$ (mm) | $40 \times 240 \times 120$ (mm) |

In this test, we used another SAM system. The new system could work with four transducers simultaneously and used two digitizers, which had two analog channels (PCI-5152, National Instruments Corporation. Austin, TX, USA). The motor control modules were similar to the old system. The new system model is described in Figure 20.

As is shown in the model of the system in Figure 20a, the sample was divided into four regions with similar areas. We performed experiments many times with different transducers, and the results are described in Figure 20c.

In the test, the trigger signals for the P/R and Digitizers 1 and 2 were completely the same. Besides that, we also used four completely identical transducers. The gain of four amplifier modules on the P/R device was adjusted differently. The trigger signals entering channels 1 and 2 of the P/R device and the digitizers were similar. This sample scan result is described in Figure 20c.

Based on the resulting image in Figure 20c, we can see the signal uniformity of four different regions of this sample. Each transducer scanned a region, so the time to scan this sample was also four times faster than using a transducer.

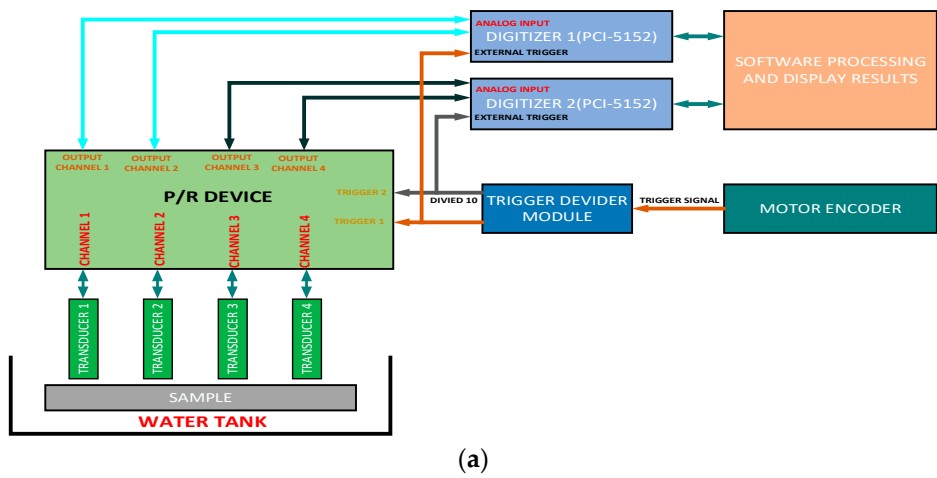

(**a**)

**Figure 20.** *Cont.*

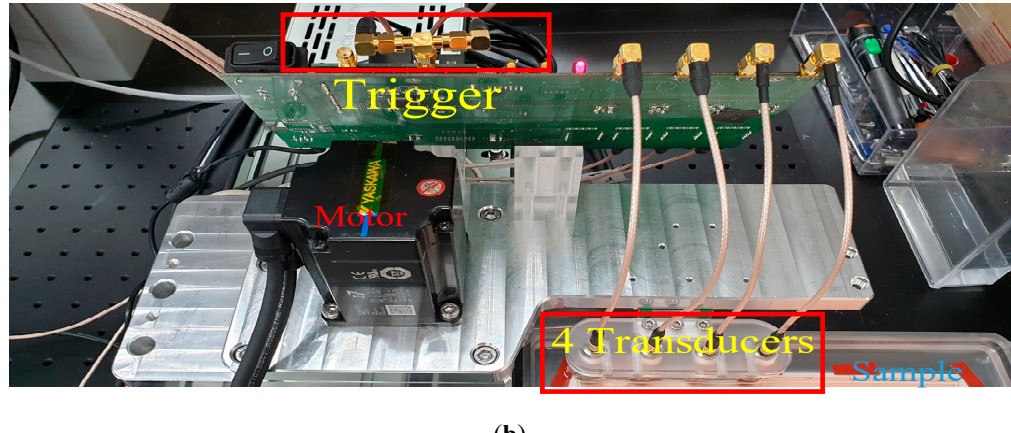

(**b**)

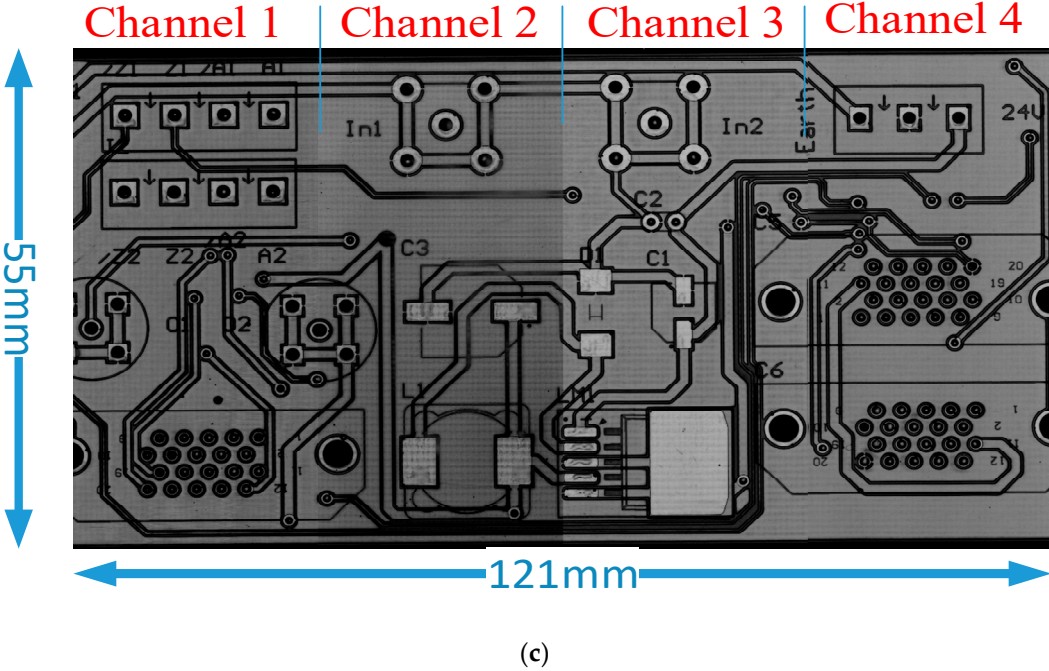

(**c**)

**Figure 20.** System scan sample with different trigger frequencies; (**a**) Schematic diagram of the system model; (**b**) The real system; (**c**) The resulting image with four different areas.

## 5. Conclusions

We proposed a multifunctional pulse generator for SAM systems with a reconfigurable and compact design. The unipolar pulse scheme could achieve a high peak amplitude, on top of the relatively high bandwidth of the device. The system implementation in a PCB scheme allows for a cost-effective and compact design. The reconfigurable structure is versatile for various applications and allows incorporation into open platforms for future research and development.

This article also presented a detailed and complete model of a P/R device used in SAM systems. The important modules required from high technology are clearly stated in the design. The P/R device is designed with four channels, which means that the speed of image scans in a SAM system can be increased four times, compared with that of a single-channel system. Independent gain adjustment channels also create a different resolution in each region. A high resolution is also important when one crucial area needs to be carefully monitored.

The techniques in PCB design with high-frequency signals are applied effectively in this design to help improve the signal and increase the bandwidth of the signal. Besides that, this paper also

makes major contributions to systematizing each of the essentials in the design of P/R equipment for SAM scanning.

One of the other outstanding contributions is a summary of the tested and statistical damping resistor values for different transducers. These have significant implications for users operating systems that use this type of equipment. Saving setup time and adjusting parameters, as well as optimizing the achieved results, help produce brighter and sharper images. Tests have shown the apparent impact of these damping resistors in enhancing the SNR of the signal echo. The echo signals from layers of the sample are also distinguished more clearly, thereby improving the resolution between layers while the system scans samples with multiple layers. The image results have increased CNRs, which increase image quality.

The design can expand to have eight channels active at the same time. This means that the SAM system will reduce the scan time of a sample by many times, compared with systems that only operate with one or two channels. This has important implications for production carried out on production lines.

**Author Contributions:** Conceptualization, supervision, and funding acquisition, J.O.; methodology, N.T.B.; software, N.T.B., T.T.N.D., T.H.V. and Q.C.B.; validation, B.-G.K. and J.O.; writing—original draft preparation, N.T.B., T.M.T.N., S.P., D.T.P., J.C. and Y.-H.K.; writing—review and editing, all authors. All authors have read and agreed to the published version of the manuscript.

**Funding:** This research was supported by the Pukyong National University Development Project Research Fund (PhiNX program), 2019.

**Conflicts of Interest:** The authors declare no conflict of interest.

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
