# Peer review of "Design of a Multichannel Pulser/Receiver and Optimized Damping Resistor for High-Frequency Transducer Applied to SAM System"

_applsci, doi:10.3390/app10238388_

Round 1

Reviewer 1 Report

The manuscript presents a multichannel pulse/receiver with a damping resistor. Following are my comments and suggestions.

1] The manuscript looks like a design report. More detailed explanation and analysis of how the authors optimized the unipolar pulser design are required. Please address performance optimization in terms of noise, bandwidth, and reconfigurability.

2] What could be the performance advantages of the proposed system over the existing ones?

3] Please provide a performance summary of the fully optimized system in a table with a comparison with the state-of-the-art systems.

4] What were the target impedance of microstrip lines and striplines? Also, by using termination techniques, how much reductions did you achieve?

5] In Fig. 17, the label of the X-axis should be “Gain.”

Author Response

Reviewer 1

The manuscript presents a multichannel pulse/receiver with a damping resistor. Following are my comments and suggestions.

1). The manuscript looks like a design report. More detailed explanation and analysis of how the authors optimized the unipolar pulser design are required. Please address performance optimization in terms of noise, bandwidth, and reconfigurability.

Thanks to reviewer's comment. This is very helpful in perfecting the manuscript

As we show in Figure 6 with the transducer impulse characteristics. The design is intended to reduce overshoot and ringing. This is done from optimizing the pulse from the FPGA to the STHV800 chip and changing the damping resistor value.

Optimization of pulses from the FPGA chip to the STHV800 was performed on the PCB by analyzing the influence of the printed circuit on the line impedance and the matching length of the lines. Also GND partition to reduce noise on printed circuit.

The target of the device is to trigger the transducer and then receive echo signal from the transducer. For high frequency transducer, trigger signal with small duty is required. The device has the ability to change the width of the pulse to match the different types of transducers to optimize the echo signal amplitude. In Figure 14, we present the trigger signal-generating block diagram, the duty change can be done easily by changing the variable value from the software.

We have changed in the manuscript (“4.1 Unipolar Pulser Generator

As we mentioned above, the pulse optimization for the probe is done in 2 parts.”)

2). What could be the performance advantages of the proposed system over the existing ones?

Thanks to reviewer's comment. This is very helpful in perfecting the manuscript

As we have shown the P/ R device used for the SAM system to scan the sample. Therefore, the time to scan the sample affects the production speed and work efficiency. Devices of big brands in the world such as UT320 (around 30,000usd) or DPR500 (over 10,000usd) are only designed for 1 or 2 channels. Moreover, the optimization of damping resistance values gives the highest efficiency with different types of transducers. The result of increasing the SNR of the received signal is as we have presented in Table 5. Our design has a much lower price than other products.

We have changed in the manuscript (“introduction The P / R device design has 4 channels of which make it easy to deploy a SAM….)

3). Please provide a performance summary of the fully optimized system in a table with a comparison with the state-of-the-art systems.

Thanks to reviewer's comment. This is very helpful in perfecting the manuscript

We have added the table 9 to the manuscript. Compare the specifications of the equipment we designed with the UT320 device (this is a new and expensive device that costs around 30,000 USD)

4). What were the target impedance of microstrip lines and striplines? Also, by using termination techniques, how much reductions did you achieve?

Thanks to reviewer's comment.

We have added Fig. 10 (c, d) jute is the impedance calculation of the microstrip and stripline in this design. All of them are matching 50Ohm.

During the design process we have used the signal integrity feature of Altinum Designer software to perform signal analysis. The values presented in the manuscript are the results of simulation and experimental measurements.

In Fig. 16 (a) We test the signal on the PCB with and without using a thenvenin resistor to limit the counter-discharge signal.

5). In Fig. 17, the label of the X-axis should be “Gain.”

Thanks to reviewer's comment. This is very helpful in perfecting the manuscript

We have changed in the manuscript.

Reviewer 2 Report

In section 1, the authors are suggested to extend the state of the art of similar SAM systems, already reported in the scientific literature, and to discuss in-depth the already reported bibliographic references not limited to the mere citing.

In section 4, the authors are suggested to further highlight the novelties of the proposed P/R system and provide comparisons with previously reported architectures, for instance in terms of scanning speed (by adding a new sub-section).

Also, the authors have to demonstrate in their designed system, the improvements obtained by employing 4-transducers instead of only one, as previously reported in the "Introduction" section.

The authors are recommended to improve the legibility of graphs reported in Figure 16a and 16b, as well as those in Figure 18c and 18d, whereas Figure 21a must be enlarged to improve its comprehensibility.

Author Response

Reviewer 2

1). In section 1, the authors are suggested to extend the state of the art of similar SAM systems, already reported in the scientific literature, and to discuss in-depth the already reported bibliographic references not limited to the mere citing.

Thanks to reviewer's comment. This is very helpful in perfecting the manuscript.

We changed in the introduction (“ The SAM system has been used in many different fields,….”)

2). In section 4, the authors are suggested to further highlight the novelties of the proposed P/ R system and provide comparisons with previously reported architectures, for instance in terms of scanning speed (by adding a new sub-section).

Thanks to reviewer's comment. We have changed in the manuscript.

We changed in the 4.5 Scanning sample with four transducers at the same time

 (“One of the highlights of our design is its,….”)  and add more table 9.

3). Also, the authors have to demonstrate in their designed system, the improvements obtained by employing 4-transducers instead of only one, as previously reported in the "Introduction" section.

Thanks to reviewer's comment. We have changed in the manuscript.

We changed in the introduction (“ The P / R device design has 4 channels of which make it easy to,….”)

4). The authors are recommended to improve the legibility of graphs reported in Figure 16a and 16b, as well as those in Figure 18c and 18d, whereas Figure 21a must be enlarged to improve its comprehensibility.

Thanks to reviewer's comment. This is very helpful in perfecting the manuscript.

We have changed in the manuscript.

Round 2

Reviewer 1 Report

The manuscript has been revised, addressing the reviewer's comments and questions. I don't have further questions.